# Location Planning of Charging Stations for Electric Buses in Public Transport Considering Vehicle Scheduling: A Variable Neighborhood Search Based Approach

Nils Olsen [†] and Natalia Kliewer *,[†]

Department of Information Systems, Freie Universität Berlin, Garystr. 21, 14195 Berlin, Germany; nils.olsen@fu-berlin.de
* Correspondence: natalia.kliewer@fu-berlin.de
† These authors contributed equally to this work.

**Abstract:** Many public transport companies have recently launched projects testing the operation of electric buses. Progressively, traditional combustion engine buses are being replaced by electric buses. In such cases, some stops on bus lines are equipped with charging technology. Combustion engine buses can operate for an entire day without having to refuel. By contrast, electric buses have considerably shorter ranges and need to recharge their batteries throughout a day. For cost-efficient use of electric buses, charging stations must be located within the road network so that required deadhead trips are as short as possible, but attention must also be paid to construction costs. In contrast to vehicle scheduling, which is a more short-term planning task of public transport companies, location planning of charging stations is a long-term planning problem and requires a simultaneous solving of both optimization problems. Specifically, location planning and vehicle scheduling have to be considered simultaneously in order to open up optimization potentials by comparison to sequential planning, since locations of charging stations directly influence the resulting vehicle rotations. To this purpose, we present a novel solution method for the simultaneous optimization of location planning of charging stations and vehicle scheduling for electric buses in public transport, using variable neighborhood search. By a computational study using real-world public transport data, we show that a simultaneous consideration of both problems is necessary because sequential planning generally leads to either infeasible vehicle rotations or to significant increases in costs. This is especially relevant for public transport companies that start operating electric bus fleets.

**Keywords:** location planning; vehicle scheduling; electric buses; charging stations; partial charging

## 1. Introduction

In the last years, awareness of climate change and sustainable operations has increased significantly throughout the entire economy and public life. Electromobility is currently considered a highly relevant technology in order to make public transport systems more sustainable and environmentally friendly. Therefore, traditional buses with combustion engines are being progressively replaced by electric buses. Electrically powered buses facilitate a locally emission-free movement which leads to minimal emission levels of greenhouse gases, dust particles, and nitrogen oxides. Seeking to improve the quality of life, especially in congested urban areas, electric buses enable much more quietly operations [1].

At present, the electric energy required for powering electric buses is either provided by batteries or is generated by fuel cells from hydrogen, methanol, or similar fuel [2]. Due to the lower energy density of modern electric batteries compared to common tank capacities for hydrogen or methanol, battery-powered buses involve the greatest challenges for bus operations. For this reason, we focus on battery electric buses (BEBs) within this work. However, the methodology and results of this work can be transferred to any other type of electric engine. We will consider electric bus and battery electric bus as synonyms.

Traditional combustion engine buses can often operate for an entire day without having to refuel. By contrast, modern BEBs have only a fraction of the ranges of combustion engine buses and need to recharge their batteries several times a day [3]. Nowadays, BEBs are charged overnight at vehicle depots after the completion of their daily operations. In addition, the vehicles are charged at charging stations during shorter waiting periods while operating (opportunity charging). Energy transmission occurs either conductively by a wire or inductively. In some cases, the vehicle batteries are also replaced with a fully charged battery (battery swapping).

With a view, for example, to the current real-world bus project at the Schiphol Airport in Amsterdam, the Netherlands, the bus company Connexxion operates with up to 100 BEBs at the present time [4]. Electric VDL Citea buses are operated within this project, with batteries capable of storing 215 kWh which results in a range between 80 and 120 km. The batteries are charged inductively with fast charging systems. Most modern electric buses like the *Irizar ie Bus* are able to store about 350 kWh and may operate up to 17 h in urban bus systems without charging [5].

In recent years, many other public transport companies have launched similar pilot projects testing the operation of BEBs. An overview on current projects is provided by [6]. Most projects initiated aim towards substituting diesel buses with BEBs during the daily services while retaining cost-minimal vehicle rotations. In such cases, charging systems are established at some stops on the bus lines to facilitate the recharging of the vehicle batteries during operation. For a cost-efficient deployment of BEBs, the charging stations must be built within the road network so that deadhead trips are as short as possible or are not necessary at all. Longer deadhead trips increase the operational costs and may lead to higher demands for buses.

Therefore, construction costs for charging stations as well as the buses' purchase and operational costs have to be considered at the planning stage. The planning process of public transport companies consists principally of strategic, tactical, and operational planning tasks, which differ with regard to the time periods considered. Figure 1 provides an overview of the planning process. Strategic planning comprises the network design and line planning. The network design determines stop points and necessary infrastructure, particularly including the distribution of charging stations within the road network. In this scope, specific technical aspects such as energy grids' transmission capacities or restrictions imposed by local conditions may be considered [7,8]. Within the tactical planning, timetables are constructed according to the previously planned lines. Operational planning determines the deployment of vehicles and personnel.

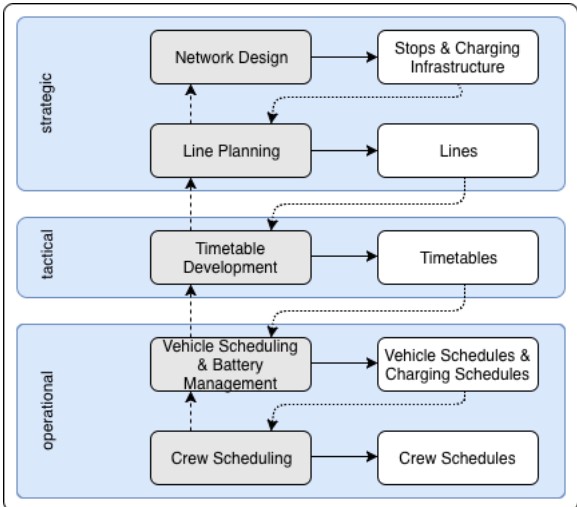

**Figure 1.** Overview of the planning process arising for companies in public transport when deploying BEBs.

The first operational planning task is vehicle scheduling, which specifies the vehicle deployment for operating service trips offered daily. Service trips denote trips to transport passengers from a departure stop via intermediate stops to an arrival stop at fixed times determined by a timetable. The objective is to assign the set of service trips to vehicles at minimum costs. As part of this task, each service trip must be covered exactly once, each vehicle must execute a feasible sequence of trips (vehicle rotation) without time overlaps, and each vehicle must start and end its rotation at the same depot. This optimization problem commonly refers to the term *Vehicle Scheduling Problem* (VSP). Between successive service trips a vehicle can perform deadhead trips without transporting passengers if necessary. If BEBs are considered within vehicle scheduling, restricted operating ranges due to limited battery capacities and battery charging must be taken into account. This extended optimization problem is commonly denoted as the *Electric Vehicle Scheduling Problem* (E-VSP). While charging, a vehicle stops at a charging station for a specific time period depending on the battery's remaining energy (State of Charge, SoC). Batteries can be either fully or partially charged. The task of determining when, where, and to what extent a battery is charged is denoted as battery management which is closely related to vehicle scheduling.

Unlike vehicle scheduling, which is a more short-term planning task in operational planning, location planning of charging stations is a long-term planning task belonging to strategic network planning and requires a simultaneous optimization of location planning of charging stations and vehicle scheduling for BEBs. Both optimization problems have to be considered simultaneously in order to open up optimization potentials by comparison to sequential planning. At the present time, there are solution approaches to the E-VSP considering fixed locations of charging stations determined in advance, on the one hand. On the other hand, location planning problems for charging stations are being solved to provide for the operation of cost-minimal vehicle rotations computed for buses without range limitations by BEBs. Both approaches belong to a sequential planning.

Simultaneous problem solving is always applicable when a public transport company fully or partially substitutes its fleet of diesel buses with BEBs for the first time. This is particularly the case because charging stations are not usually available within public transport systems yet and need to be built. Furthermore, it is expected that in the future private energy companies will operate networks of charging stations, especially within urban areas, that can be used by vehicles and buses. Some of these networks already exist, such as *E.on Drive* in Germany, but it is expected that such offers will be expanded in the future [9]. In this scenario, each transport company has to pay a usage fee in order to charge a vehicle at specific stations. While location planning of charging stations is a long-term planning problem, vehicle scheduling is carried out every time the timetable changes. However, the simultaneous approach is still applicable because then it is based on the modified timetable and the set of charging stations provided by the energy companies. The construction costs for building a charging station then correspond to the usage fees.

In this paper, we present a novel solution method for the simultaneous optimization of location planning of charging stations and vehicle scheduling for BEBs in public transport to open up potentials for cost savings in comparison with a sequential planning. To do so, we develop a solution approach based on *Variable Neighborhood Search* (VNS), which has been successfully applied to real-world combinatorial optimization problems in a variety of application areas [10]. We propose a heuristic solution approach because the E-VSP and the location planning problem are both difficult to solve, especially with regard to larger instances. Following Lenstra and Rinnooy [11] and Yang and Sun [12], both problems are NP-hard. Simultaneous problem solving is expected to be no less difficult [13]. Within our solution approach we incorporate complete as well as partial charging procedures of the vehicle batteries. By a computational study, we prove the need for simultaneous optimization as opposed to sequential planning. We show that simultaneous problem solving is necessary because sequential planning generally leads to either infeasible vehicle

rotations or to significant increases in costs. Further on, we discern that the incorporation of partial charging procedures leads in principle to major cost savings.

This paper is outlined as follows: In Section 2 we provide an overview of existing work about scheduling of electric vehicles and location planning of charging stations for BEBs. In Section 3 we define the problem to be solved formally. Following this, we introduce the metaheuristic solution method in Section 4. In Section 5 we perform comprehensive computational experiments and analyze the results in order to make key statements. We provide conclusions and present potentials for further research in Section 6.

## 2. Literature Overview

In this section, we give an overview of related work. As mentioned above, existing work can generally be divided into scheduling of BEBs assuming fixed locations of charging stations and location planning of charging stations for given vehicle rotations. Consequently, we begin by discussing existing solution approaches for scheduling BEBs in public transport. We then present literature on location planning of charging stations.

### 2.1. Scheduling Electric Buses

As one of the first contributions dealing with alternative engine types within vehicle scheduling, Stasko and Gao [14] present a solution method for the VSP taking into account different engine options. The solution approach is based on integer programming. Engines powered by compressed natural gas (CNG) are considered besides combustion engines. The approach aims at reducing emission levels within vehicle scheduling.

Reuer et al. [15] consider a mixed fleet of vehicles consisting of electrically powered buses and buses without range limitations within the basic VSP. The authors apply a time-space network based exact solution method for the VSP introduced by Kliewer et al. [16] to solve the enhanced optimization problem. Solutions obtained to this problem contain optimal flow values through the network. Therefore, strategies for flow decomposition are necessary to obtain vehicle rotations. The authors analyze six strategies for flow decomposition that aim at maximizing the proportion of feasible vehicle rotations for BEBs. Battery charging is assumed to be performed within constant time periods. The authors show that a simple substitution of traditional buses with BEBs leads to widely infeasible vehicle schedules.

Haghani and Banihashemi [17] consider a fleet consisting entirely of range restricted vehicles. They consider vehicle scheduling with route and time constraints in order to limit the lengths and durations of vehicle rotations. However, battery charging is not considered. The authors propose one exact and two heuristic solution models together with techniques for reducing the problem sizes in order to solve even larger-scale problem instances. Chao and Xiaohong [18] consider battery swapping in addition to limited operating ranges of BEBs within the VSP. To solve the problem, a solution method based on a Non-dominated Sorting Genetic Algorithm (NSGA-II) is introduced. A case study based on real-world data taken from a project in Shanghai is performed to analyze the solution approach. Li [19] addresses vehicle scheduling of BEBs with either battery swapping or charging and presents a model for restricting the maximum route distance. Both fast charging and battery swapping are presumed to be performed within constant time windows, but the time for fast charging depends on the location. Adler and Mirchandani [20] deal with scheduling of BEBs incorporating charging procedures at given charging stations located within the road network. To solve the problem, they present a column-generation approach. A heuristic method is presented to obtain necessary initial solution. The algorithm is based on a greedy algorithm and computes vehicle rotations under consideration of range limitations and charging. In this work, again full chargings of vehicle batteries are assumed.

As one of the first authors, Wen et al. [21] address the E-VSP with partial chargings. They present an exact solution method based on mixed integer programming and an adaptive large neighborhood search heuristic approach. The results demonstrate that the exact solution methods is only applicable to small problem instances. However, the

heurstic solution approach also solves larger instances in a reasonable amount of time. van Kooten Niekerk et al. [22] also consider partial charging procedures of BEBs. The authors introduce a solution approach based on column generation. Charging times depend linearly on a battery's SoC. Furthermore, battery aging and time-dependent energy prices are considered. The authors show that in some cases, the consideration of partial charging procedures leads to cost savings.

Recently, Wang et al. [23] proposed an exact solution method for the E-VSP based on dynamic programming. Within this contribution, battery aging is particularly considered. The objective of the solution method is minimize the total costs especially incorporating costs for battery replacements during the life spans of the vehicles deployed. By a computational study, the authors analyze the influence of different working loads, battery management, and working temperatures of batteries on resulting vehicle schedules.

### 2.2. Location Planning of Charging Stations for Electric Buses

At the present time, only few publications deal with location planning of charging stations for BEBs in public transport. Kunith et al. [2] present a mixed integer linear optimization model for determining locations for charging stations for a bus route. The model is based on a set covering problem. The objective is to minimize the number of charging stations needed. The authors consider constraints imposed by the buses' operation and the battery charging process. In addition, different energy consumption scenarios are considered to reflect external influencing factors on the buses' energy consumption, such as traffic volume and weather conditions. Standard optimization libraries are used for solving the problem.

Berthold et al. [24] propose a mixed integer linear program in order to determine optimal locations of charging stations for the electrification of a single bus line in Mannheim. The problem is solved by using standard optimization libraries. Furthermore, partial charging procedures and battery aging effects over several time periods are considered. Since the problem is very complex, the solution approach is not suitable for larger instances. Xyliaa et al. [25] develop a dynamic optimization model to establish a charging infrastructure for BEBs in Stockholm, Sweden, considering restricted waiting times at intermediate stops on service trips given by the schedule and different currents of the charging systems imposed by local conditions. They provide statements about the application possibilities of BEBs in urban areas and effects on vehicle rotations. Within both works, no line changes of the buses used are considered.

Liu et al. [26] consider energy consumption uncertainties within location planning of charging stations for BEBs in public transport. Therefore, the authors propose a robust optimization model represented by a mixed integer linear program. Using real-world data, the authors show that the proposed solution model can provide optimal locations for charging stations that are robust against uncertain energy consumption of BEBs. Lin et al. [27] introduce a spatial-temporal model for a large-scale planning of charging-stations for BEBs in public transport. The authors consider characteristics of BEBs operation and plug-in fast charging technologies. The model is represented by a mixed-integer second-order cone programming formulation with high computational efficiency. A case study using data from Shenzhen, China is used to analyse the robustness of the solution model to timetable changes.

Based on the solution method presented in this paper, Stumpe et al. [13] present an exact mathematical model for integrated optimization of vehicle scheduling with BEBs and location planning for charging stations. The authors particularly perform a robustness analysis and study the impact of technological aspects such as battery capacity, charging power, and energy consumption as well as economic issues containing investment costs for charging stations and electric buses. A computational study points out that the exact solution model introduced is not capable of solving realistic problem instances to optimality.

Regarding related optimization problems in the scope of transportation, there are some contributions dealing with the charging infrastructure for electric vehicles. Regarding

*Vehicle Routing Problems* (VRP) with electric vehicles, Worley et al. [28] propose a solution approach for the simultaneous determination of optimal locations for charging stations and vehicle routes. They show that this approach leads to lower total costs of the vehicle deployment by comparison to locations of charging stations known a priori. Schiffer and Walther [29] also deal with the simultaneous determination of locations for charging stations and routes for electric vehicles. The authors extend this optimization problem by considering uncertain characteristics of the customers to be served. Uncertain spatial customer distributions, demand, and service time windows are particularly addressed. The authors introduce a robust optimization approach based on adaptive large neighborhood search. Vehicle routing comprises different challenges and conditions than vehicle scheduling and therefore needs other solution approaches. Consequently, it is not possible to draw concrete statements with regard to the E-VSP.

### 2.3. Summary and Need for Further Research

Table 1 presents the main characteristics of the presented literature. As described there, there is no existing work that deals with scheduling of BEBs and location planning of charging stations simultaneously. However, as underlined by Worley et al. [28] with regard to vehicle routing, a simultaneous optimization opens up potentials for cost savings. It is to be expected that a simultaneous problem solving will also be beneficial for scheduling of BEBs in public transport. In addition, partial charging procedures have not yet been considered sufficiently within the scope of scheduling BEBs. As shown by van Kooten Niekerk et al. [22] for fixed locations of charging stations, the incorporation of partial charging procedures facilitates further optimization potentials. Simultaneous problem solving under consideration of partial charging procedures forms the basic idea of our contribution.

**Table 1.** Overview of the main characteristics of related literature.

| Reference | E-VSP | E-VRP | Mixed veh.fleet | Electric veh.fleet | w/o Line Changes | Fixed chrg.stat. | Fixed veh.rot. | Partial Charging |
|---|---|---|---|---|---|---|---|---|
| Stasko and Gao [14] | ● | | ● | | | ● | ● | |
| Haghani and Banihashemi [17] | ● | | | ● | | ● | | |
| Worley et al. [28] | | ● | | ● | | | | |
| Chao and Xiaohong [18] | ● | | | ● | | ● | | |
| Li [19] | ● | | | ● | | ● | | |
| Reuer et al. [15] | ● | | ● | | | ● | ● | |
| Adler and Mirchandani [20] | ● | | | ● | | ● | | |
| Wen et al. [21] | ● | | | ● | | ● | | ● |
| Berthold et al. [24] | ● | | | ● | ● | | ● | ● |
| van Kooten Niekerk et al. [22] | ● | | | ● | | ● | | ● |
| Xyliaa et al. [25] | ● | | | ● | ● | | ● | ● |
| Liu et al. [26] | ● | | | ● | | | ● | ● |
| Schiffer and Walther [29] | | ● | | ● | | | | ● |
| Lin et al. [27] | ● | | | ● | | | ● | ● |
| Wang et al. [23] | ● | | | ● | | ● | | ● |
| Stumpe et al. [13] | ● | | | ● | | | | ● |

### 3. Problem Description and Cost Model

In this section, we present the *Electric Vehicle Scheduling Problem with Location Planning of Charging Stations* (E-VSP-LP) as the key problem being solved in this paper. In the following, we first introduce the parameters of the problem. Afterwards, we introduce decision variables and the objective function.

We assume a public transportation network given by a set $S = \{s_1, \ldots, s_n\}$ of $n \in \mathbb{N}$ stop points also containing the set of vehicle depots $D \subseteq S$. Service trips are defined by a given timetable as a set $T = \{t_1, \ldots, t_m\}$ with $m \in \mathbb{N}$. A service trip $t \in T$ is characterized by its departure and arrival time as well as its departure and arrival stop. For any pair $(s_i, s_j) \in S \times S$ of stop points there is a specific distance and travel time that can be different depending on whether the trip is a service or deadhead trip. In our study, we do not consider opportunity charging of BEBs during the execution of service trips. Consequently, the set $S$ contains the departure and arrival stop of each service trip $t \in T$ as well as the set of depots. The aim is to assign the service trips contained in $T$ to a set of BEBs that are substantially determined by their battery capacities. There may be other specifications such as vehicle dimensions or passenger capacities. Each combination of these features is denoted as a *vehicle type*. To recharge the vehicle batteries, charging stations can be built at each stop point of $S$. The installed charging system at a charging station considerably influences the time needed for charging. A vehicle can be either fully or partially charged, which also affects the charging time.

For the deployment of a BEB fixed costs $c_{fixed}^{bus} > 0$ incure independently of the executed trips. Each charging or trip operated during a vehicle rotation results in operational costs. Therefore, we consider time costs per hour $c_{time}^{bus} > 0$ and for the distances covered of $c_{distance}^{bus} > 0$. The equipment of stop points with charging technology causes fixed costs $c_{fixed}^{charging} > 0$. These costs may be different, depending on the type of the charging system to be installed or the location. For instance, it is more expensive to build a charging station at a busy crossing than in a quiet side street.

We define decision variables $y_s \in \{0, 1\}$, $\forall s \in S$ and $x_v \in \{0, 1\}$, $\forall v \in V$ denoting the decision whether a charging station is built at stop point $s$ or respectively, whether a vehicle $v$ is used or not. The objective of the simultaneous optimization problem is to minimize the total costs for a given timetable and potential locations of charging stations. Accordingly, fixed costs for BEBs as well as charging stations and operational costs for the buses' operation must be minimized. The objective function can be formulated as

$$min \quad \underbrace{\sum_{s \in S} c_{fixed}^{charging} \cdot y_s}_{location\ planning} + \underbrace{\sum_{v \in V} c_{fixed}^{bus} \cdot x_v}_{vehicle\ costs} + \underbrace{\sum_{v \in V} \sum_{t \in v} \left( c_{time}^{bus} \cdot dur(t) + c_{distance}^{bus} \cdot len(t) \right)}_{operative\ costs}. \quad (1)$$

$$\underbrace{\phantom{\sum_{v \in V} \sum_{t \in v} \left( c_{time}^{bus} \cdot dur(t) + c_{distance}^{bus} \cdot len(t) \right)}}_{vehiclescheduling}$$

A trip's duration is specified by $dur(t) \geq 0$ and a trip's length by $len(t) \geq 0$. The objective function's value may be interpreted as the total costs caused by a first investment into an electrification of a public transport company's fleet and infrastructure for a specific timetable period. Variable costs for the maintenance of the charging infrastructure or battery replacements are not considered within this work.

In this paper, we solve the E-VSP-LP heuristically as large real-world instances cannot be solved to optimality in an acceptable time [13]. For that reason, we do not present a formal model at this point. However, we refer to Stumpe et al. [13] for a comprehensive mathematical problem formulation and further insights.

## 4. A Variable Neighborhood Search Based Solution Method for the E-VSP-LP

In this section, we discuss our solution approach for the E-VSP-LP. The objective is to find vehicle rotations for BEBs and locations for charging stations simultaneously and at a minimum cost. We begin by presenting the basic procedure of our heuristic solution method. The solution method consists primarily of generating initial solutions first and then finding new solutions with lower total costs. To do so, we introduce a savings algorithm for generating initial solutions in Section 4.2. Afterwards, we present an algorithm for improvement based on VNS in Section 4.3.

### 4.1. General Approach

Algorithm 1 provides the main procedure of our solution method. The set of scheduled service trips to be assigned and an initial set of charging stations, together with their locations, serve as the input data. Already existing charging infrastructure, for example due to the implementation of previous pilot projects, may be included in the set of charging stations. Usually, at the beginning of the algorithm the set of charging stations is empty. The algorithm basically consists of two consecutive steps: First, we use a savings algorithm to generate initial sets of vehicle rotations for BEBs and charging stations (l. 1). Subsequently, we use this initial solution as the input for an improvement method based on VNS, which we denote as BVNS (l. 2). The algorithm terminates by returning the best solution found. The two key Algorithms 2 and 3 are explained in the following sections.

---
**Algorithm 1** Main Variable Neighborhood Search.

---
    **Input:** scheduled service trips $T$, charging stations $S$
    **Output:** vehicle rotations $V$, charging stations $\overline{S}$

  1: $(V', S') \leftarrow \texttt{SA}(T, S)$;
  2: $(V'', S'') \leftarrow \texttt{BVNS}(V', S')$;
  3: **return** $V'', S''$;

---

### 4.2. Savings Algorithm for Generating Initial Solutions

The savings algorithm was first introduced by Clarke and Wright [30] to solve VRPs heuristically. The objective of vehicle routing is to determine an optimal set of routes seeking to service a number of customers with a fleet of vehicles. Following Cordeau et al. [31], the savings algorithm is one of the most commonly used methods for vehicle routing in practice. Starting from routes each containing one customer the basic procedure is to compute cost savings iteratively for merging two routes into the same one. Within each iteration the merging that results in the highest saving is performed. A saving consists of fixed and operative costs saved. This procedure terminates when no further mergings can be performed. While this algorithm has been applied generally to VRPs, we adapt this algorithm hereinafter in order to apply the same procedure to the E-VSP-LP.

Algorithm 2 shows the procedure for generating initial solutions to the E-VSP-LP formally using the idea of cost savings. The set of scheduled service trips to be assigned and an initial set of charging stations, together with their locations, serve as the input data. The algorithm begins by adding a vehicle rotation for each scheduled service trip, now containing only the associated trip together with a deadhead trip from and to the depot (l. 4). If these vehicle rotations are not feasible for BEBs the entire optimization problem is infeasible. Within each iteration of the algorithm those two vehicle rotations (l. 7 and 8) are merged that lead to a feasible rotation and entail the highest saving. Therefore, the set of service trips of both rotations to be merged are processed consecutively, in order of departure times (l. 9). Since the SoC mostly influences the feasibility of a vehicle rotation besides temporal restrictions the algorithm aims at adding charging procedures as often as possible. For this purpose, starting with a new and empty vehicle rotation (l. 10), four different cases are considered for each service trip of the rotations to be merged. First, we check whether a charging procedure can be performed at an existing charging station of $S$ before executing the current service trip, taking into account necessary deadhead trips (l. 12). If this can be done, necessary deadhead trips, the charging procedure, and the service trip are added (l. 13). If this is not possible, we examine whether the current service trip can be executed without detours to charging stations (l. 14). If the SoC is insufficient, we check whether the current service trip can be executed by building a new charging station at the trip's departure stop and performing a charging procedure (l. 16). Lastly, the same is checked but for the latest position of the vehicle, which is less strict because the deadhead trip is executed after the charging procedure (l. 18). If none of these options can be carried

out, the current merging is aborted (l. 20). When a merging is feasible, the saving for merging two vehicle rotations $v, w \in V$ into a new rotation $\overline{v, w}$ is given by

$$s(v, w) = c_{fix}^{bus} - \delta \cdot c_{fix}^{charging} - (o(\overline{v, w}) - o(v) - o(w)) \tag{2}$$

where $o(v) \geq 0$, $\forall v \in V$ denotes the operational costs for each vehicle rotation and $\delta \in \mathbb{N}$ the number of additionally respectively fewer needed charging stations. After each iteration the merging is performed that involves the highest positive saving (l. 25). Then, the set $S$ of charging stations is modified, the new vehicle rotation is added, and the rotations merged are removed (l. 26 and 27). If no positive savings exist, the algorithm terminates and returns the sets of vehicle rotations and charging stations (l. 29). Hence, solutions generated by this procedure are always feasible.

The procedure of Algorithm 2 is based on the heuristic solution method proposed for the E-VSP by Adler and Mirchandani [20]. Within this algorithm, the charging stations are assumed to be known a priori and cannot be changed. However, within Algorithm 2, we extend the procedure from Adler and Mirchandani [20] significantly by incorporating location planning for charging stations.

---

**Algorithm 2** Savings Algorithm (SA).

---

　　**Input:** scheduled service trips $T$, charging stations $S$
　　**Output:** vehicle rotations $V$, charging stations $\overline{S}$

　1: $V \leftarrow \varnothing$
　2: $\overline{S} \leftarrow S$
　3: **for all** $t \in T$ **do**
　4: 　　Add a vehicle rotation to $V$ containing only $t$;
　5: **end for**
　6: **while** TRUE **do**
　7: 　　**for all** $v \in V$ **do**
　8: 　　　　**for all** $w \in V \setminus \{v\}$ **do**
　9: 　　　　　　Determine the set $\overline{T}$ of service trips of $v \cup w$;
　10: 　　　　　　Create a new vehicle rotation $\overline{v}$ without trips;
　11: 　　　　　　**for all** $t \in \overline{T}$ **do**
　12: 　　　　　　　　**if** $\overline{v}$ can be recharged at an existing charging station and execute $t$ **then**
　13: 　　　　　　　　　　Add necessary deadhead trips, charging, $t$ to $\overline{v}$;
　14: 　　　　　　　　**else if** $\overline{v}$ can execute $t$ **then**
　15: 　　　　　　　　　　Add necessary deadhead trips, $t$ to $\overline{v}$;
　16: 　　　　　　　　**else if** $\overline{v}$ can be recharged at the departure stop of $t$ and execute $t$ **then**
　17: 　　　　　　　　　　Add charging station to $\overline{S}$, necessary deadhead trips, charging, $t$ to $\overline{v}$;
　18: 　　　　　　　　**else if** $\overline{v}$ can be recharged at its current position and execute $t$ **then**
　19: 　　　　　　　　　　Add charging station to $\overline{S}$, charging, necessary deadhead trips, $t$ to $\overline{v}$;
　20: 　　　　　　　　**else break**;
　21: 　　　　　　　　**end if**
　22: 　　　　　　**end for**
　23: 　　　　**end for**
　24: 　　**end for**
　25: 　　Make move with the highest saving $s(v, w)$;
　26: 　　Remove rotations $v$ and $w$ from $V$; Add $\overline{v}$ to $V$;
　27: 　　Add new charging stations to $\overline{S}$;
　28: 　　**if** No positive savings exist **then**
　29: 　　　　**return** $V, \overline{S}$;
　30: 　　**end if**
　31: **end while**

---

### 4.3. Variable Neighborhood Search for Improvement

To finding new solutions with lower total costs, we use a VNS based solution method. VNS was first introduced by Hansen et al. [10]. Solution approaches based on VNS have already been extensively considered in the literature and have been proven to be suitable for numerous practical problems with realistic data sizes [32]. The underlying concept of VNS is a systematic change of neighborhoods, both in an improvement phase to find a local optimum and in a perturbation phase to escape from local optima. In the perturbation phase, a so-called shaking method is applied, which exerts a stochastic influence on an incumbent solution by performing stochastic changes. Even this procedure can cause a deterioration in the objective function value it has used to escape from local optima. In the improvement phase, a local search method is used to find new solutions with lower total costs.

Adapting the basic VNS concept to solve the E-VSP-LP thus requires the definition of a problem specific neighborhood structure and methods for shaking, a local search, and changing the neighborhood. Algorithm 3 provides the procedure for our solution method. The algorithm follows the *basic* VNS adapted from Hansen et al. [33]. Note that the following procedure is applicable not only for solutions generated by Algorithm 2 but also for every possible feasible solution.

---

**Algorithm 3** Basic Variable Neighborhood Search (BVNS).

---

**Input:** vehicle rotations $V$, charging stations $S$, $t_{max}$, $k_{max}$
**Output:** vehicle rotations $V$, charging stations $S$

1: $t \leftarrow 0$
2: **while** $t < t_{max}$ **do**
3:     $k \leftarrow 1$;
4:     **while** $k \leq k_{max}$ **do**
5:         $(V', S') \leftarrow \texttt{SHAKE}(V, S, k)$;
6:         $(V'', S'') \leftarrow \texttt{BESTIMPROVEMENT}(V', S', k)$;
7:         $(V, S, k) \leftarrow \texttt{NEIGHBORHOODCHANGE}((V, S), (V'', S''), k)$;
8:     **end while**
9:     $t \leftarrow CPUTIME()$;
10: **end while**
11: **return** $(V, S)$;

---

We first define a neighborhood $N_k$ of size $k \in \mathbb{N}$ by selecting $k$ vehicle rotations. The choice of the vehicle rotations will be made randomly from the entire set in order to incorporate stochastic influences. It follows the maximum neighborhood size $k_{max} \in \mathbb{N}$ as the number of vehicles used within the incumbent solution. After each iteration of shaking and local search, a neighborhood change is performed. In this step, the objective function values of the incumbent and improved solution are compared. If the improved solution is better than the incumbent, it is accepted and the size of the neighborhood is reset to the smallest possible value. Otherwise, the size of the neighborhood is increased and the procedure is repeated. The procedure terminates when the maximum computational time is exceeded. Algorithm 4 shows the procedure formally.

---

**Algorithm 4** NEIGHBORHOODCHANGE.

---

> **Input:** solutions $(V, S)$, $(V', S')$, neighborhood size $k$, objective function $f$
> **Output:** solution $(V, S)$, neighborhood size $k$

1: **if** $f(V', S') < f(V, S)$ **then**
2:     $(V, S) \leftarrow (V', S')$;
3:     $k \leftarrow 1$;
4: **else** $k \leftarrow k + 1$;
5: **end if**
6: **return** $(V, S)$, $k$;

---

Second, we use Algorithm 5 as the local search method within Algorithm 3 for improving a solution. As the total costs of a solution consist of operational costs for deadheading as well as fixed costs for vehicles and charging stations, Algorithm 5 combines the three following Algorithms 6–8, each aiming towards reducing one cost component. In Algorithm 5, the move is performed that involves the highest cost saving.

---

**Algorithm 5** BESTIMPROVEMENT.

---

> **Input:** neighborhood $N_k$, objective function $f$
> **Output:** neighborhood $N_k$

1: **return** $\min_f \{EXST(N_k), SST(N_k), SCP(N_k)\}$;

---

Algorithm 6 is used to reduce operational costs for deadheading by exchanging service trips between different vehicle rotations of a corresponding neighborhood. Therefore, a saving is computed for each pair of service trips for the neighborhood's set of vehicles that can be exchanged, and the move with the highest saving is returned.

---

**Algorithm 6** Exchange of Service Trips (EXST).

---

> **Input:** neighborhood $N_k$
> **Output:** neighborhood $N_k$

1: **for all** $v \in V$ **do**
2:     **for all** $w \in V \setminus \{v\}$ **do**
3:         **for all** $t_v \in v$ **do**
4:             **for all** $t_w \in w$ **do**
5:                 **if** $t_v$ and $t_w$ can be exchanged **then**
6:                     Compute saving;
7:                 **end if**
8:             **end for**
9:         **end for**
10:     **end for**
11: **end for**
12: Perform exchange with the highest saving;
13: **return** $N_k$;

---

Algorithm 7 aims at inserting service trips of vehicle rotations with a lower number of service trips into vehicle rotations with a higher number of service trips, again based on a neighborhood. If an insertion is possible, a saving is computed containing proportionate fixed costs for the remaining service trips, fixed costs for additional charging stations, and operational costs for possible detours. Again, the best move found is returned. The algorithm attempts to omit vehicle rotations whereby no service trips are being executed any more.

---

**Algorithm 7** Shift Service Trips (SST).

---

**Input:** neighborhood $N_k$, fixed costs for an BEB $c_{fix}^{bus}$
**Output:** neighborhood $N_k$

---

1: **for all** $v \in V$ **do**
2:      **for all** $w \in V : |ST_w| < |ST_v|$ **do**
3:          **for all** $t_w \in w$ **do**
4:              **if** $t_w$ can be inserted in $v$ **then**
5:                  Compute saving $(c_{fix}^{bus}/|ST_w|)$ less the costs for newly built charging stations
6:                  and additional operational costs;
7:              **end if**
8:          **end for**
9:      **end for**
10: **end for**
11: Perform move with the highest saving, omit a vehicle if no trips are being performed;
12: **return** $N_k$;

---

Algorithm 8 aims at decreasing the number of charging stations used by moving charging procedures from less frequented charging stations to higher frequented charging stations, considering the vehicle rotations of a neighborhood. The move is returned that is feasible and entails the highest saving including proportionate fixed costs for remaining charging procedures at a specific charging station and operational costs for additional detours. Similar to Algorithm 7, this procedure aims at omitting charging stations where chargings are no longer being performed at a specific stop point.

---

**Algorithm 8** Shift Charging Procedures (SCP).

---

**Input:** neighborhood $N_k$, charging stations $S$
**Output:** neighborhood $N_k$, charging stations $S$

---

1: Sort $S$ by the number of charging procedures performed within the entire set of vehicle rotations in ascending order;
2: **for** $s = 1$ **to** $|S| - 1$ **do**
3:      **for all** $t = |S|$ **to** $s + 1$ **do**
4:          **if** A charging of a vehicle in $N_k$ is performed at $s$ and can be shifted to $t$ **then**
5:              Compute saving $(c_{fix}^{charging}/|CP_s|)$ less additional operational costs;
6:          **end if**
7:      **end for**
8: **end for**
9: Perform move with the highest saving, omit a charging stations if no chargings are being performed;
10: **return** $N_k$, $S$;

---

While stochastic influences on incumbent solutions are already incorporated by the random selection of a neighborhood's set of vehicles, the Algorithm 9 is applied additionally within Algorithm 3. This approach is intended to enable more stochastic changes to the procedure aiming to escape from local optima. Shaking is based on the procedures given by Algorithms 6–8. Within each method call of Algorithm 9, one of the three algorithms is randomly applied if the corresponding move is feasible. This is done even though the objective function value is being worsened.

---

**Algorithm 9** `SHAKE`.

---

    **Input:** neighborhood $N_k$
    **Output:** neighborhood $N_k$

 1: Choose `EXST`, `SST` or `SCP` as $f$ at random;
 2: **if** $f(N_k)$ is feasible **then**
 3:     **return** $f(N_k)$;
 4: **else** Go to 1
 5: **end if**

---

### 4.4. Inserting Partial Chargings

In our computational study, which follows this section, we incorporate complete and partial charging procedures. So far, the algorithms presented operate with any kind of charging procedures. However, we need more algorithmic effort in order to incorporate partial chargings within Algorithms 2 and 3. To that purpose, we consider the following Algorithm 10 by Olsen and Kliewer [34]. It is applied to each vehicle rotation that is generated respectively modified within the solution procedure. As a result, Algorithm 10 either returns the set of partial charging procedures that have to be inserted into the corresponding vehicle rotation or its infeasibility. Only if a resulting vehicle rotation is feasible is it taken into further consideration.

Algorithm 10 checks iteratively, after each trip of a rotation, whether the SoC has been violated (l. 2). If this is the case, the previous trips are considered (l. 3). Each trip that begins or ends at a charging station represents a charging opportunity (l. 5). If no such possibilities are found the vehicle rotations is infeasible (l. 9). Over all charging possibilities determined, the one performed at the most highly frequented charging station is processed (l. 11). This aims at reducing the number of charging stations by shifting charging procedures from less to more highly frequented charging stations. In the next step, the vehicle rotation is divided at the specific charging station into two sub-rotations containing the previous and subsequent trips. Then, both sub-rotations are processed by the algorithm. In the case that all sub-rotations are feasible, the algorithm terminates (l. 13). If a charging station is no longer needed it is omitted. If at least one sub-rotation is infeasible, the next charging opportunity is processed (l. 15 and l.16).

---

**Algorithm 10** Inserting Partial Chargings (PCP).

**Input:** vehicle rotation $v$, set $S$ of charging stations
**Output:** vehicle rotation $v$, feasibility or infeasibility of $v$

---

 1: **for all** $t_1 \in v$ **do**
 2:     **if** SoC after executing $t_1$ is not sufficient **then**
 3:         **for all** $t_2 \in v$ previous to $t_1$ **do**
 4:             **if** Departure stop is a charging station **then**
 5:                 Save charging opportunity;
 6:             **end if**
 7:         **end for**
 8:         **if** Set of charging opportunity is empty **then**
 9:             **return** $v$, **infeasible**;
10:         **end if**
11:         Add charging opportunity at the highest frequented charging station;
12:         **if** Vehicle rotation can be performed **then**
13:             **return** $v$, **feasible**;
14:         **else**
15:             Exclude charging opportunity from the set of all opportunities;
16:             Go to 8;
17:         **end if**
18:     **end if**
19: **end for**

---

## 5. Computational Analysis

In the following, we perform our computational experiments. We first present the instances to be solved and the problem parameters. Then, we look at the results of a sequential planning approach. In this case, location planning of charging stations and vehicle scheduling of BEBs are solved one by one. Therefore, our analysis is twofold: First, we discuss the results of solving a location planning problem for charging stations to enable the operation of given cost-optimal vehicle rotations computed for traditional buses without the range limitations of BEBs. Second, we present the results of solving an E-VSP given the locations of charging stations computed in the previous step. Last, we analyze the results of simultaneous problem solving using our heuristic solution method provided by Algorithm 3 for the E-VSP-LP and compare the results to the sequential planning approaches. We specifically investigate the impact of considering complete and partial charging procedures on solutions.

### 5.1. Experimental Design

Our computational experiments are performed on 10 real-world instances that are inspired by real-world public transport data. The instances are characterized by different numbers of stop points and service trips as well as different distributions of service trips over a day. To simplify the analysis, the instances' labels reflect the numbers of service trips and stop points. The instances' distributions of cumulative service trips over the day are presented by Figure 2. The figure shows that the instances differ substantially with regard to the distributions. It is worth mentioning that the last five instances consist of subsets of the service trips taken from instance t3067_s209 for runtime reasons. In the case of instances t1580_s209 and t1487_s209 the original set of service trips was halved randomly, and in the case of instances t1060_s209, t1074_s209, and t933_s209 the set was divided into three parts also in a random way.

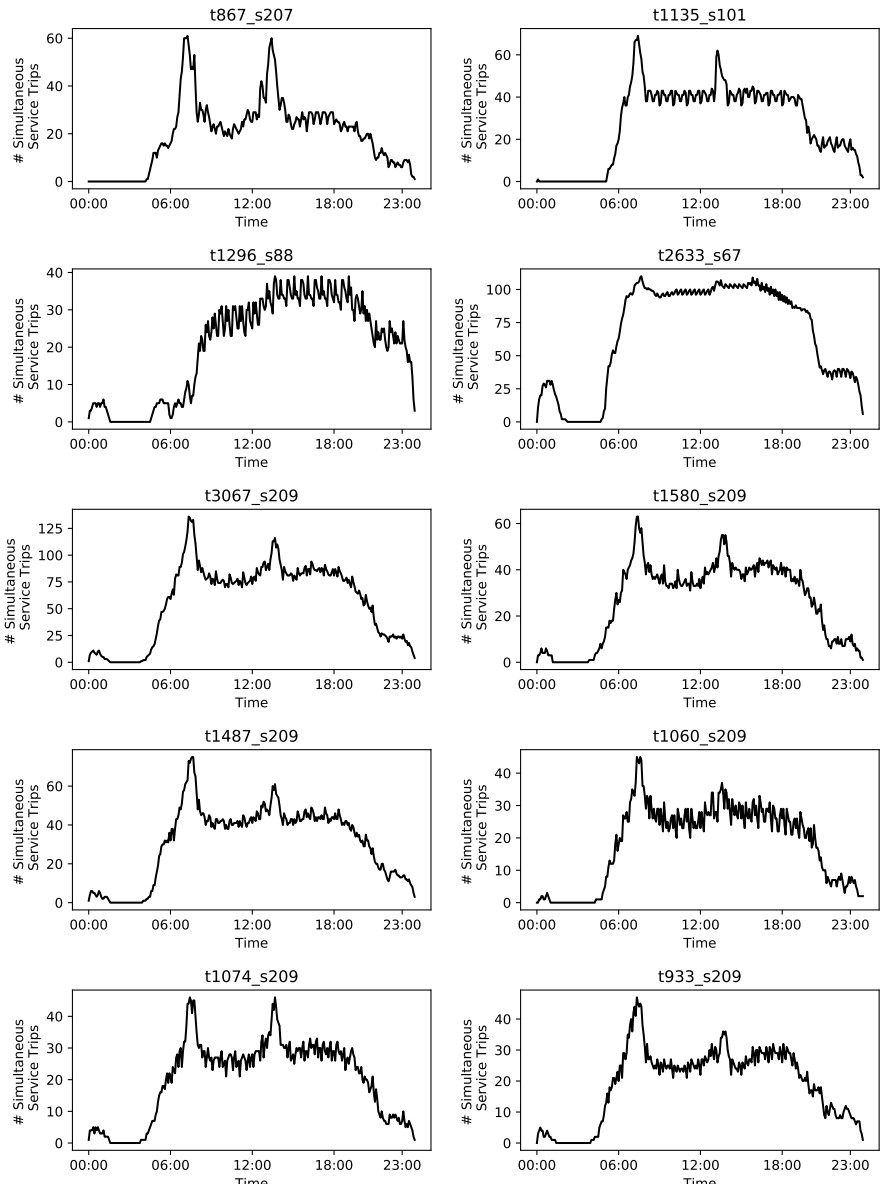

**Figure 2.** Profiles of cumulative service trips.

Within our experiments, we presume a single vehicle depot, a single vehicle type, and a single charging system. Accordingly, each timetabled service trip can be executed by every available BEB. Additionally, each BEB can charge its battery at every charging station. With regard to the practical implementations of BEBs, we assume that three buses at most can be charged at a charging station at the same time. This is because building sites for charging systems are usually restricted, especially in urban areas. In our study, we distinguish between complete and partial charging procedures. In order to incorporate battery aging, we presume that a battery's SoC ranges between 20% and 80% of a battery's capacity as indicated by Fernandez et al. [35]. In our experiments, we first presume that a vehicle is always charged up to a SoC of 80%. After that, we consider partial chargings. In that regard, the threshold until a battery is charged may vary depending on the idle times at charging stations. Irrespective of the threshold until a battery is charged during its rotation, we assume that a vehicle always begins its rotation with a fully charged battery.

Following Stamati and Bauer [36], charging modern batteries is a nonlinear and therefore complex procedure. The current during a charging process is of particular importance. As demonstrated by Olsen and Kliewer [34], the current decreases quickly

when a battery is charged to over 80% of its capacity. Below this threshold, the current is almost constant. For that reason, we assume a constant current and thus linear charging times for vehicle batteries within this paper. We assume that 5 kWh can be transferred into a vehicle battery per minute. In our study we consider chargings before the start or after the end of service trips. To reflect the lower consumption of BEBs on deadhead trips we therefore assume a consumption of 1.5 kWh per kilometer and of 1.8 kWh per kilometer driving on service trips. These parameters are inspired by the data of the previously introduced project at the Schiphol Airport in Amsterdam. At present, there is a wide range of battery capacities offered on the market that range between approximately 60 and 300 kWh. Based on this, we consider different battery capacities of 60, 120, 300 and 500 kWh within our experiments. A battery capacity of 500 kWh may be considered as a future development of battery research. Since we consider only one vehicle type at the same time, we conduct our study for each capacity. Based on Stamati and Bauer [36], a BEB in use and equipped with a 60-kWh battery causes fixed costs of about 350,000 monetary units. Measured by the battery sizes this results to fixed costs for the other vehicles of 365,000, 405,000, and 450,000 monetary units. With regard to the operational costs, we presume 0.5 units per driven kilometer and 50 units per hour of operation. Again based on the bus project in Amsterdam, the equipment of a stop point with charging technology is incorporated with fixed costs of 200,000 monetary units. We use the term "monetary units" here since we assume that these units are roughly comparable—at least in terms of scale—and, based on this, that monetary units form a system of imputed cost components.

*5.2. Location Planning of Charging Stations for the Electrification of Cost-Minimal Vehicle Rotations, Computed without Range Limitations*

We begin our computational analysis by discussing the results of solving a location planning problem for charging stations for the electrification of given cost-minimal vehicle rotations computed without range limitations. The vehicle rotations were generated using the exact optimization method for the traditional VSP by Kliewer et al. [16], which is based on a time-space network. In order to enable the operation of these rotations by BEBs, charging stations are added to the network and charging procedures are inserted into the vehicle rotations. Partial charging procedures are performed, since the idle times at potential charging stations are given by the vehicle rotations. The objective is to maximize the proportion of vehicle rotations that are feasible for BEBs. Ideally, this procedure should ensure the holistic operation of the timetabled service trips by BEBs. For this purpose, we adapt the location planning problem for charging stations introduced by Berthold et al. [24] and solve it using standard optimization libraries.

Table 2 provides the results of solving a location planning problem for charging stations, containing the proportions and absolute numbers of feasible vehicle rotations for BEBs together with the numbers and proportions of charging stations needed for each instance and each battery capacity. Additionally, the optimal number of vehicles used is indicated when no range limitations are considered. If the totality of all vehicle rotations is feasible for BEBs, the operational and total costs are specified for subsequent analyses. First, we observe that in the vast majority of cases the holistic electrification of vehicle rotations by means of inserting charging procedures is not possible. It is apparent that this observation holds regardless of the instance to be solved. However, the proportion of feasible vehicle rotations grows with increasing battery capacities. We can observe that every instance can be entirely served by BEBs in the case of a battery capacity of 500 kWh. In some cases, this situation already occurs with a battery capacity of 300 kWh and in a single case with 120 kWh. However, none of the instances can be entirely served by BEBs with a battery capacity of 60 kWh. Regarding a battery capacity of 60 kWh, the proportions of feasible vehicle rotations fluctuates widely and ranges between 7.25% and 79.63%. With regard to charging stations, it can be concluded that the numbers of stop points equipped with charging technology decreases significantly when the battery capacities grow. Instance t1296_s88 shows the biggest reduction in the number of charging stations needed from

48.86% to 6.81%. The operational costs of feasible vehicle rotations decrease slightly when the battery capacities grow, which can be attributed to fewer charging procedures required.

In summary, a sequential planning solving at first a standard VSP without incorporating the special features of BEBs and subsequently a location planning problem for charging stations is generally insufficient, leading to widely infeasible solutions. This approach is only suitable if the ranges of BEBs rise sharply in the future. The results obtained serve as lower bounds for the numbers of BEBs used and as an upper bound for the numbers of charging stations needed in the evaluation of the simultaneous solution approach.

**Table 2.** Results of solving a location planning problem for charging stations for the electrification of cost-minimal vehicle rotations computed without range restrictions incorporating partial charging procedure.

| Instance | Battery Capacity (kWh) | # Vehicles | # Stations | Operat. Costs (Mio) | Feasible Rotations | Total Costs (Mio) |
|---|---|---|---|---|---|---|
| t876_s207 | 60 | 69 | 47 (22.71%) | - | 5 (7.25%) | - |
| | 120 | 69 | 44 (21.25%) | - | 31 (44.93%) | - |
| | 300 | 69 | 33 (15.94%) | - | 62 (89.86%) | - |
| | 500 | 69 | 7 (3.38%) | 1.127 | 69 (100%) | 33.93 |
| t1135_s101 | 60 | 75 | 33 (32.67%) | - | 43 (57.33%) | - |
| | 120 | 75 | 27 (26.73%) | - | 69 (92%) | - |
| | 300 | 75 | 15 (14.85%) | 1.351 | 75 (100%) | 35.48 |
| | 500 | 75 | 2 (1.98%) | 1.349 | 75 (100%) | 35.60 |
| t1296_s88 | 60 | 47 | 43 (48.86%) | - | 28 (59.68%) | - |
| | 120 | 47 | 32 (36.36%) | - | 42 (80.37%) | - |
| | 300 | 47 | 25 (28.40%) | - | 42 (80.37%) | - |
| | 500 | 47 | 6 (6.81%) | 0.114 | 47 (100%) | 22.76 |
| t2633_s67 | 60 | 125 | 29 (43.28%) | - | 74 (58.4%) | - |
| | 120 | 125 | 21 (32.34%) | - | 80 (64%) | - |
| | 300 | 125 | 17 (25.37%) | - | 117 (93.6%) | - |
| | 500 | 125 | 8 (11.94%) | 2.652 | 125 (100%) | 60.91 |
| t3067_s209 | 60 | 165 | 90 (43.06%) | - | 88 (53.33%) | - |
| | 120 | 165 | 69 (33.01%) | - | 154 (93.33%) | - |
| | 300 | 165 | 39 (18.66%) | - | 162 (96.97%) | - |
| | 500 | 165 | 14 (6.69%) | 3.045 | 165 (100%) | 80.79 |
| t1580_s209 | 60 | 75 | 55 (26.31%) | - | 39 (52%) | - |
| | 120 | 75 | 45 (21.53%) | - | 61 (81.33%) | - |
| | 300 | 75 | 20 (9.56%) | 1.342 | 75 (100%) | 36.71 |
| | 500 | 75 | 7 (3.34%) | 1.319 | 75 (100%) | 36.82 |
| t1487_s209 | 60 | 89 | 53 (25.35%) | - | 46 (51.79%) | - |
| | 120 | 89 | 37 (17.71%) | - | 87 (97.76%) | - |
| | 300 | 89 | 24 (11.48%) | 1.696 | 89 (100%) | 43.74 |
| | 500 | 89 | 7 (3.34%) | 1.672 | 89 (100%) | 43.47 |
| t1060_s209 | 60 | 54 | 43 (20.57%) | - | 43 (79.63%) | - |
| | 120 | 54 | 30 (14.35%) | 0.988 | 54 (100%) | 28.20 |
| | 300 | 54 | 13 (6.22%) | 0.987 | 54 (100%) | 26.11 |
| | 500 | 54 | 3 (1.43%) | 0.985 | 54 (100%) | 26.04 |
| t1074_s209 | 60 | 56 | 39 (18.66%) | - | 31 (55.36%) | - |
| | 120 | 56 | 33 (15.78%) | - | 52 (92.86%) | - |
| | 300 | 56 | 16 (7.65%) | 0.986 | 56 (100%) | 27.67 |
| | 500 | 56 | 6 (2.87%) | 0.985 | 56 (100%) | 27.69 |
| t933_s209 | 60 | 54 | 35 (16.74%) | - | 23 (42.60%) | - |
| | 120 | 54 | 25 (11.96%) | - | 32 (77.78%) | - |
| | 300 | 54 | 15 (7.17%) | 0.971 | 54 (100%) | 26.59 |
| | 500 | 54 | 4 (1.91%) | 0.963 | 54 (100%) | 26.26 |

### 5.3. Scheduling of Electric Buses Given Fixed Locations of Charging Stations

We now discuss the results of solving an E-VSP with given locations of charging stations. The set of charging stations determined by the previous experiment within Section 5.2 serves as the input, since this set is already optimal if corresponding solutions are feasible for BEBs. Following Section 1, the objective of the E-VSP is to minimize the

number of buses in use and the operational costs for deadheading while covering each service trip. In order to ensure comparability, partial chargings are performed. Because the E-VSP is NP-hard and exact solution methods are not suitable for solving large real-world instances in general, as in our experiments, we solve the E-VSP heuristically here.

To do so, we use our main solution method from Algorithm 1 in a reduced version. Within both Algorithms 2 and 3, which represent the main components of Algorithm 1, we disable modifications of the charging stations. Within Algorithm 2, we only allow the assignment of service trips to vehicles without charging or with detours to existing charging stations. The other two cases are omitted. Within Algorithm 3, we modify the Algorithms 5 and 9 by disabling Algorithm 8 within each procedure. This approach means that the set of charging stations cannot change in this experiment. While the following results are not necessarily optimal due to the heuristic solving, we show that they provide reasonable bounds for our analysis within the next section.

An overview of the results of this experiment is given by Table 3, providing the numbers of vehicles used as well as operational and total costs. The number of charging stations is taken from the previous experiment. In contrast to that, now each solution is feasible, which was to be expected because of the constraints imposed by the E-VSP. Consequently, the total costs are specified for each instance and each battery capacity, containing fixed costs for buses used and charging stations as well as operational costs. First of all, the results show that in most cases where feasible vehicle rotations were computed in the first experiment described in Section 5.2, the solving of an E-VSP provides similar results regarding the numbers of vehicles used and total costs. In some cases, the number of vehicles required is slightly higher than in the first experiment, which can be traced back to the heuristic solution approach. Furthermore, the operational costs are marginally increased. However, the solutions of this experiment converge towards the optimal solutions and thus provide a reasonable benchmark for subsequent analyses. Regarding the numbers of vehicles used, one can observe that the fewer the proportions of feasible vehicle rotations determined within the first experiment, the more vehicles are required when solving the E-VSP. This is understandable because the closely-timed service trips of the vehicle rotations when no range limitations are considered do not provide enough time for rechargings. This leads to an increasing demand for vehicles. For example, considering instance t1580_s209, the optimal numbers of vehicles used is obtained for battery capacities of 500 kWh and 300 kWh. As the proportion of feasible vehicle rotations reduces rapidly for 120 kWh and 60 kWh within the first experiment (81.33% respectively 52%), the need for additional vehicles rises significantly (6 respectively 12 additional vehicles). Regarding the operational costs, we note that higher demands for vehicles generally leads to decreasing operational costs. This is because less deadhead trips and chargings have to be performed due to the shorter rotations.

In conclusion, solving an E-VSP with given locations of charging stations always leads to feasible vehicle rotations, which is in contrast to the first experiment. However, this solution approach generally entails increases in costs due to additional deadhead trips, likely leading to increasing demands for vehicles. The results obtained serve as upper bounds for the analysis of the simultaneous problem solving to be conducted in the following section.

**Table 3.** Results of solving an E-VSP given locations of charging stations by reduced Algorithm 1 incorporating partial charging procedures.

| Instance | Battery Capacity (kWh) | # Vehicles | # Stations | Operat. Costs (Mio) | Total Costs (Mio) |
|---|---|---|---|---|---|
| t876_s207 | 60 | 80 (+10) | 47 | 1.019 | 40.77 |
| | 120 | 75 (+6) | 44 | 1.060 | 39.72 |
| | 300 | 72 (+3) | 33 | 1.098 | 38.51 |
| | 500 | 69 (+0) | 7 | 1.157 | 33.96 |
| t1135_s101 | 60 | 87 (+12) | 33 | 1.187 | 39.89 |
| | 120 | 82 (+7) | 27 | 1.219 | 37.90 |
| | 300 | 76 (+1) | 15 | 1.265 | 35.79 |
| | 500 | 75 (+0) | 2 | 1.379 | 35.62 |
| t1296_s88 | 60 | 61 (+14) | 43 | 0.082 | 32.18 |
| | 120 | 49 (+3) | 32 | 0.112 | 28.16 |
| | 300 | 49 (+3) | 25 | 0.108 | 25.79 |
| | 500 | 47 (+0) | 6 | 0.116 | 22.77 |
| t2633_s67 | 60 | 144 (+19) | 29 | 2.193 | 59.84 |
| | 120 | 137 (+12) | 21 | 2.438 | 57.69 |
| | 300 | 131 (+6) | 17 | 2.514 | 59.82 |
| | 500 | 126 (+1) | 8 | 2.621 | 61.32 |
| t3067_s209 | 60 | 179 (+14) | 90 | 2.681 | 87.83 |
| | 120 | 171 (+6) | 69 | 2.820 | 82.49 |
| | 300 | 169 (+4) | 39 | 2.871 | 81.07 |
| | 500 | 166 (+1) | 14 | 2.994 | 81.19 |
| t1580_s209 | 60 | 87 (+12) | 55 | 1.133 | 45.33 |
| | 120 | 81 (+6) | 45 | 1.289 | 42.10 |
| | 300 | 75 (+0) | 20 | 1.367 | 36.74 |
| | 500 | 75 (+0) | 7 | 1.323 | 36.83 |
| t1487_s209 | 60 | 101 (+12) | 53 | 1.421 | 50.02 |
| | 120 | 92 (+3) | 37 | 1.573 | 44.40 |
| | 300 | 90 (+1) | 24 | 1.682 | 44.13 |
| | 500 | 89 (+0) | 7 | 1.753 | 43.53 |
| t1060_s209 | 60 | 59 (+5) | 43 | 0.952 | 32.35 |
| | 120 | 55 (+1) | 30 | 0.991 | 28.56 |
| | 300 | 54 (+0) | 13 | 0.989 | 26.11 |
| | 500 | 54 (+0) | 3 | 0.986 | 26.04 |
| t1074_s20 | 60 | 64 (+8) | 39 | 0.897 | 33.05 |
| | 120 | 59 (+3) | 33 | 0.971 | 30.76 |
| | 300 | 57 (+1) | 16 | 0.988 | 28.07 |
| | 500 | 56 (+0) | 6 | 0.994 | 27.69 |
| t933_s209 | 60 | 64 (+10) | 35 | 0.956 | 32.06 |
| | 120 | 58 (+4) | 25 | 0.961 | 28.38 |
| | 300 | 55 (+1) | 15 | 0.970 | 26.99 |
| | 500 | 54 (+0) | 4 | 0.969 | 26.27 |

### 5.4. Simultaneous Optimization of Vehicle Scheduling and Charging Infrastructure

We now discuss the results of simultaneous optimization of scheduling of BEBs and location planning for charging stations, i.e., solving the E-VSP-LP, using our solution method given by Algorithm 1. We begin by presenting the results obtained by Algorithm 2 for generating initial solutions. Then, we discuss the results of Algorithm 3 for finding new solutions with lower total costs. In this experiment we consider complete as well as partial charging procedures in order to enable a comparison with the previous experiments. We conclude this chapter by a runtime analysis.

### 5.4.1. Summary of Results for Generating Initial Solutions

Table 4 provides the results of using Algorithm 2 for generating initial solutions containing feasible sets of vehicle rotations and charging stations. The results contain the total and operational costs as well as the numbers of buses and charging stations used for each instance and each battery capacity. Additionally, the differences in the total costs

are specified when enabling partial charging procedures, and the best solutions found are in bold.

**Table 4.** Results of Algorithm 2 for generating initial vehicle rotations for electric buses and locations for charging stations considering complete and partial charging procedures.

| Instance | Battery Cap.(kWh) | Complete Chargings | | | | Partial Chargings | | | |
|---|---|---|---|---|---|---|---|---|---|
| | | # Vehicles | # Stations | Operat. Costs (Mio) | Tot. Costs Costs (Mio) | # Vehicles | # Stations | Operat. | Tot. Costs |
| t876_s207 | 60 | 90 | 2 | 1.620 | 33.62 | 86 | 2 | 1.621 | 32.22 (−1.40) |
| | 120 | 76 | 1 | 1.392 | 29.38 | 75 | 1 | 1.397 | 29.02 (−0.36) |
| | 300 | 76 | 1 | 1.322 | 32.35 | 75 | 1 | 1.322 | 31.94 (−0.40) |
| | 500 | 76 | 1 | 1.307 | 35.75 | 75 | 1 | 1.307 | 35.31 (−0.45) |
| t1135_s101 | 60 | 107 | 1 | 1.990 | 39.69 | 104 | 2 | 1.991 | 38.89 (−0.80) |
| | 120 | 94 | 2 | 1.644 | 36.45 | 92 | 3 | 1.644 | 35.97 (−0.48) |
| | 300 | 91 | 1 | 1.528 | 38.63 | 89 | 2 | 1.529 | 38.07 (−0.56) |
| | 500 | 80 | 1 | 1.493 | 37.74 | 79 | 1 | 1.501 | 37.30 (−0.44) |
| t1296_s88 | 60 | 86 | 6 | 0.729 | 32.33 | 82 | 7 | 0.730 | 31.18 (−**1.15**) |
| | 120 | 74 | 3 | 0.487 | 28.25 | 71 | 4 | 0.489 | 27.40 (−0.84) |
| | 300 | 64 | 1 | 0.408 | 26.58 | 62 | 1 | 0.412 | 25.77 (−0.81) |
| | 500 | 58 | 1 | 0.384 | 26.73 | 56 | 1 | 0.391 | 25.84 (−0.89) |
| t2633_s67 | 60 | 148 | 18 | 3.818 | 60.11 | 151 | 16 | 3.709 | 60.56 (+0.44) |
| | 120 | 144 | 16 | 3.307 | 59.87 | 146 | 14 | 3.292 | 60.08 (+0.21) |
| | 300 | 139 | 12 | 2.978 | 62.27 | 141 | 11 | 2.787 | 62.64 (+0.37) |
| | 500 | 134 | 6 | 2.892 | 64.69 | 135 | 5 | 2.815 | 64.82 (+0.12) |
| t3067_s209 | 60 | 182 | 50 | 3.618 | 79.81 | 180 | 46 | 3.621 | 78.12 (−1.70) |
| | 120 | 178 | 48 | 3.346 | 80.31 | 175 | 43 | 3.378 | 78.00 (−2.31) |
| | 300 | 174 | 36 | 3.114 | 82.58 | 171 | 33 | 3.164 | 80.67 (−1.91) |
| | 500 | 171 | 12 | 3.087 | 83.03 | 169 | 12 | 3.096 | 82.14 (−0.89) |
| t1580_s209 | 60 | 108 | 5 | 1.966 | 41.01 | 109 | 2 | 1.721 | 40.37 (−0.65) |
| | 120 | 98 | 1 | 1.601 | 37.62 | 98 | 1 | 1.583 | 37.60 (−0.02) |
| | 300 | 91 | 1 | 1.474 | 38.58 | 91 | 1 | 1.462 | 38.57 (−0.01) |
| | 500 | 87 | 1 | 1.287 | 40.68 | 87 | 1 | 1.276 | 40.67 (−0.01) |
| t1487_s209 | 60 | 124 | 4 | 2.464 | 46.86 | 118 | 4 | 2.464 | 44.76 (−2.10) |
| | 120 | 102 | 1 | 1.940 | 39.42 | 99 | 2 | 1.940 | 38.58 (−0.85) |
| | 300 | 102 | 1 | 1.797 | 43.36 | 98 | 2 | 1.792 | 41.98 (−1.38) |
| | 500 | 98 | 1 | 1.752 | 46.10 | 95 | 2 | 1.751 | 45.01 (−1.10) |
| t1060_s209 | 60 | 82 | 2 | 1.490 | 30.69 | 78 | 3 | 1.493 | 29.54 (−1.15) |
| | 120 | 66 | 2 | 1.218 | 25.81 | 64 | 3 | 1.219 | 25.33 (−0.48) |
| | 300 | 63 | 1 | 1.132 | 26.89 | 61 | 2 | 1.134 | 26.34 (−0.56) |
| | 500 | 60 | 1 | 1.112 | 28.36 | 57 | 2 | 1.121 | 27.27 (−1.09) |
| t1074_s209 | 60 | 86 | 2 | 1.496 | 32.09 | 85 | 5 | 1.499 | 32.49 (+0.40) |
| | 120 | 72 | 1 | 1.216 | 27.74 | 71 | 4 | 1.218 | 28.13 (+0.39) |
| | 300 | 72 | 1 | 1.132 | 30.54 | 71 | 3 | 1.194 | 30.69 (+0.16) |
| | 500 | 67 | 1 | 1.105 | 31.50 | 66 | 3 | 1.184 | 31.63 (+0.13) |
| t933_s209 | 60 | 81 | 6 | 1.527 | 31.37 | 82 | 7 | 1.498 | 31.95 (+0.57) |
| | 120 | 66 | 1 | 1.171 | 25.51 | 67 | 2 | 1.169 | 26.12 (+0.61) |
| | 300 | 65 | 1 | 1.081 | 27.66 | 66 | 2 | 1.089 | 28.32 (+0.66) |
| | 500 | 60 | 1 | 1.044 | 28.29 | 61 | 2 | 1.075 | 29.03 (+0.73) |

We first compare the results to the first experiment conducted and described in Section 5.2. We observe that in two of the 17 cases, when the first experiment lead to feasible vehicle rotations, the total costs obtained by the application of the savings algorithm were already lower by comparison to solving a location planning problem for charging stations. In the other cases, higher total costs are obtained. In general, the higher total costs arise from higher demands for vehicles needed within the savings algorithm. Regarding each instance, the numbers of vehicles used has increased, which is reasonable due to the heuristic solution procedure of the savings algorithm. The solving of instance *t*1296_*s*88 leads to the highest increase of 23.4%. By contrast, the number of charging stations used decreases in every case. In some cases, such as instance *t*1060_*s*209, the number of charging stations needed is enormously reduced (30 to two). However, since the costs for additional vehicles prevail over the cost savings arising from the lower number of charging stations used, the total costs increase. This holds true both for complete and partial charging procedures. Regarding these two charging procedures, the total costs obtained are lower in seven of the ten instances for all battery capacities when partial

charging procedures are enabled. On average, total cost savings of about 1.2% are achieved. Only in three cases are the total costs higher when considering partial chargings.

We now compare the initial solutions with the results obtained and described in Section 5.3. With regard to the total costs, our observations are twofold: In those cases in which the solving of a location planning problem led to infeasible vehicle rotations, the application of Algorithm 2 leads to lower total costs by comparison to the results obtained by solving an E-VSP. In the other cases where feasible solutions were obtained, the total costs are higher, arising from a higher demand for vehicles needed as indicated previously. Basically, the results computed by Algorithm 2 merely serve as the input for improvement methods and thus do not serve as the final results. For this reason, the clarified statements are not particularly significant. In the next section, we present the results of improvement using our solution approach based on VNS.

5.4.2. Summary of Results for Variable Neighborhood Search for Improvement

In order to carry out a final comparison between sequential planning and simultaneous problem solving, we now present the results of our solution method given by Algorithm 3 for finding new solutions with lower total costs. We use the initial solutions presented in the previous section as the input data. Table 5 shows the results, containing numbers of vehicles and charging stations used, as well as operational and total costs for each instance and each battery capacity. Additionally, the differences in the total costs are specified when enabling partial charging procedures, and the best solutions found are in bold.

Again, we first compare the results to solving a location planning problem for charging stations at given vehicle rotations. In those cases, where feasible solutions were computed and shown in Section 5.2, the total costs obtained by applying Algorithm 3 are almost of the same quality. In some cases, the total costs are slightly higher, which is most likely due to the heuristic solving. However, in certain scenarios, even better solutions are achieved which can be explained by the utilization of the degrees of freedom. Simultaneous problem solving enables shorter and fewer deadhead trips to charging stations, leading to lower operational and fixed costs for vehicles. This effect would be intensified if exact solution methods were used. As the sequential planning approach leads mostly to infeasible solutions, the simultaneous problem solving is generally preferable.

We now discuss the results with regard to solving an E-VSP with given locations of charging stations as carried out and described in Section 5.3. The most significant observation is that the total costs obtained by the simultaneous problem solving are always below the results of solving an E-VSP with fixed charging stations. This holds true for each combination of instance and battery capacity. The primary reasons for this are that the VNS based approach leads either to the same or slightly higher numbers of vehicles. Similarly, considerably lower numbers of charging stations needed are achieved due to the simultaneous solution procedure, leading to significant cost savings. Additionally, the operational costs are reduced for the most part, which can be explained by the shorter deadhead trips to charging stations. As the cost savings exceed the increased costs for additional vehicles, the solutions generated entail significantly lower total costs. It is interesting to oberserve that the greatest costs savings are achieved for instances that contain peak times of cumulative service trips over the day. This can be explained by the fact that peak times of service trips over the day allow the vehicles to recharge their batteries during times with reduced offers. In conclusion, simultaneous problem solving enables significant cost savings and is always preferable to solving an E-VSP with given locations of charging stations.

**Table 5.** Results of Algorithm 3 containing vehicle schedules for electric buses and charging infrastructure after 100.000 iterations considering complete and partial charging procedures.

| Instance | Battery Cap.(kWh) | Complete Chargings | | | | Partial Chargings | | | |
|---|---|---|---|---|---|---|---|---|---|
| | | # Vehicles | # Stations | Operat. Costs (Mio) | Tot. Costs Costs (Mio) | # Vehicles | # Stations | Operat. | Tot. Costs |
| t876_s207 | 60 | 79 | 6 | 1.317 | 30.46 | 77 | 4 | 1.348 | 29.29 (−1.17) |
| | 120 | 76 | 3 | 1.334 | 30.57 | 75 | 2 | 1.392 | 29.26 (−1.31) |
| | 300 | 74 | 3 | 1.317 | 32.03 | 73 | 2 | 1.322 | 31.38 (−0.65) |
| | 500 | 73 | 2 | 1.254 | 34.60 | 72 | 1 | 1.277 | 33.92 (−0.68) |
| t1135_s101 | 60 | 86 | 31 | 1.592 | 39.44 | 85 | 30 | 1.617 | 38.86 (−0.58) |
| | 120 | 81 | 22 | 1.512 | 36.57 | 79 | 20 | 1.555 | 35.38 (−1.19) |
| | 300 | 77 | 13 | 1.617 | 36.05 | 76 | 13 | 1.656 | 35.68 (−0.37) |
| | 500 | 76 | 2 | 1.267 | 35.96 | 75 | 2 | 1.293 | 35.54 (−0.42) |
| t1296_s88 | 60 | 58 | 37 | 0.089 | 26.63 | 56 | 32 | 0.089 | 27.68 (−1.95) |
| | 120 | 49 | 24 | 0.112 | 23.99 | 49 | 21 | 0.112 | 23.24 (−0.75) |
| | 300 | 49 | 21 | 0.108 | 25.20 | 48 | 20 | 0.110 | 24.55 (−0.65) |
| | 500 | 48 | 9 | 0.112 | 23.96 | 48 | 7 | 0.113 | 23.46 (−0.50) |
| t2633_s67 | 60 | 139 | 21 | 2.217 | 56.11 | 138 | 19 | 2.219 | 55.26 (−0.85) |
| | 120 | 136 | 18 | 2.445 | 56.58 | 135 | 16 | 2.450 | 55.72 (−0.86) |
| | 300 | 130 | 16 | 2.528 | 59.17 | 129 | 14 | 2.534 | 58.27 (−0.90) |
| | 500 | 128 | 7 | 1.609 | 61.95 | 127 | 6 | 1.617 | 61.26 (−0.69) |
| t3067_s209 | 60 | 182 | 48 | 2.627 | 78.32 | 177 | 36 | 2.694 | 73.64 (−4.68) |
| | 120 | 172 | 37 | 2.796 | 74.82 | 170 | 27 | 2.809 | 71.60 (−3.22) |
| | 300 | 171 | 26 | 2.854 | 78.60 | 169 | 18 | 2.819 | 75.76 (−2.84) |
| | 500 | 169 | 12 | 2.894 | 81.94 | 167 | 11 | 2.937 | 80.83 (−1.11) |
| t1580_s209 | 60 | 80 | 41 | 1.698 | 39.94 | 79 | 39 | 1.706 | 39.10 (−0.84) |
| | 120 | 79 | 41 | 1.751 | 40.83 | 78 | 36 | 1.754 | 39.22 (−1.61) |
| | 300 | 77 | 14 | 1.318 | 36.00 | 76 | 12 | 1.324 | 35.10 (−0.90) |
| | 500 | 75 | 8 | 1.317 | 37.06 | 75 | 7 | 1.318 | 36.81 (−0.25) |
| t1487_s209 | 60 | 99 | 38 | 1.448 | 45.59 | 96 | 31 | 1.451 | 42.81 (−2.80) |
| | 120 | 92 | 31 | 1.567 | 42.89 | 91 | 24 | 1.569 | 40.78 (−2.11) |
| | 300 | 92 | 23 | 1.534 | 44.54 | 90 | 19 | 1.561 | 42.76 (−1.78) |
| | 500 | 90 | 6 | 1.494 | 43.49 | 89 | 6 | 1.533 | 43.08 (−0.41) |
| t1060_s209 | 60 | 59 | 37 | 0.951 | 30.85 | 57 | 31 | 0.958 | 28.65 (−2.19) |
| | 120 | 56 | 30 | 0.982 | 28.92 | 56 | 27 | 0.983 | 28.17 (−0.75) |
| | 300 | 55 | 15 | 0.983 | 27.00 | 54 | 13 | 0.988 | 26.10 (−0.90) |
| | 500 | 55 | 2 | 0.984 | 26.23 | 54 | 3 | 0.985 | 26.03 (−0.20) |
| t1074_s209 | 60 | 64 | 26 | 0.913 | 29.81 | 62 | 23 | 0.912 | 28.36 (−1.45) |
| | 120 | 59 | 19 | 0.963 | 27.24 | 57 | 19 | 0.968 | 26.52 (−0.73) |
| | 300 | 57 | 14 | 0.981 | 27.56 | 56 | 16 | 0.981 | 27.66 (+0.00) |
| | 500 | 56 | 7 | 0.982 | 27.93 | 56 | 4 | 0.983 | 27.18 (−0.75) |
| t933_s209 | 60 | 61 | 27 | 0.939 | 29.03 | 60 | 24 | 0.941 | 27.94 (−1.10) |
| | 120 | 58 | 23 | 0.948 | 27.86 | 56 | 19 | 0.951 | 26.14 (−1.73) |
| | 300 | 55 | 15 | 0.964 | 26.98 | 54 | 15 | 0.970 | 26.59 (−0.40) |
| | 500 | 55 | 4 | 0.959 | 26.70 | 54 | 4 | 0.962 | 26.26 (−0.45) |

Lastly, we investigate the impact of enabling partial charging procedures within vehicle rotations. The results clearly specify that the incorporation of partial chargings is more realistic and opens up optimization potentials. The number of vehicles as well as charging stations used is lower in almost all cases. This leads to significant cost savings up to 4.68% compared to the best solution found for one of the two sequential approaches. On average, savings of 1.17% over all instances and battery capacities can be observed. The same total costs are achieved in only one case. Furthermore, the more vehicles are used, the higher the cost savings are. For this reason, the cost savings generally decrease when the battery capacities increase.

It is worth noting that the clarified statements would also hold true for exact solution methods for the E-VSP-LP. Exact solving would even strengthen the results because of the expected lower total costs. Figure 3 illustrates the key statements made within this chapter. The figure provides an overview of the total costs obtained by the different solution approaches presented for the instances t1060_s209, t1135_s101, and t3067_s209 and for all battery capacities. The instances are chosen among all instances presented since they cover characteristic problem sizes and distributions of cumulative service trips over the day. Comparable behavior is to be expected for instances with similar characteristics not shown here. It is important to note that the total costs are only specified for feasible solutions.

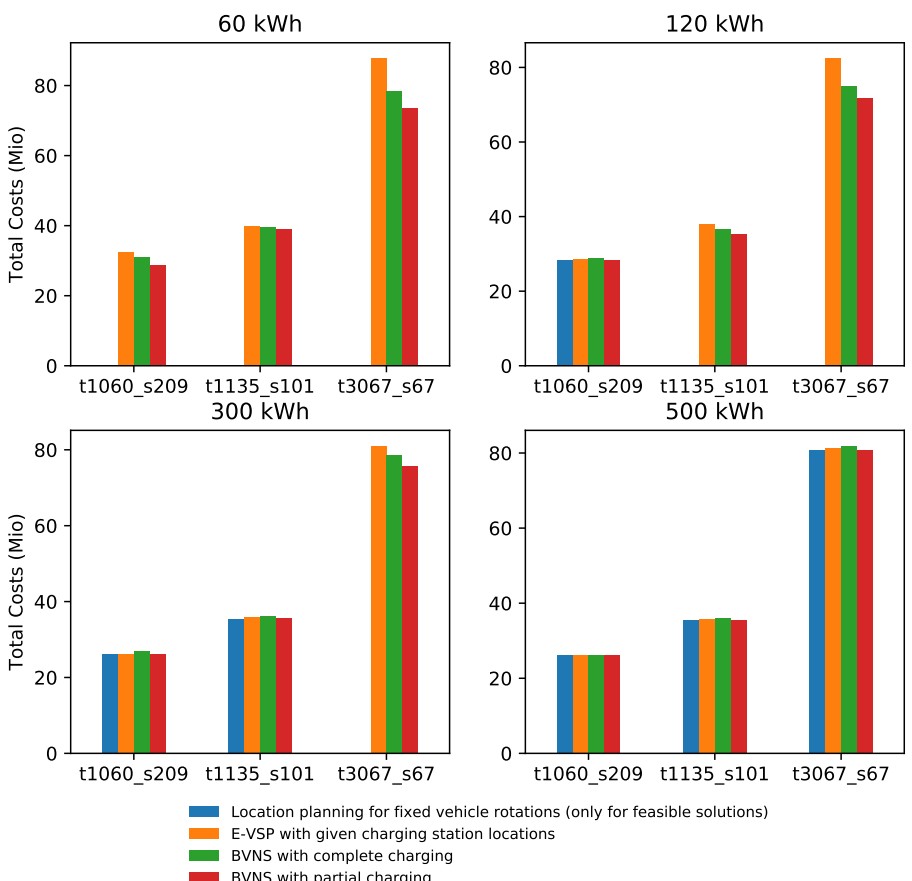

**Figure 3.** Overview of total costs obtained by the different solution approaches presented for all battery capacities.

### 5.4.3. Convergence Analysis

The experiments were performed on a common desktop computer (Intel(R) Core(TM) i7-6700 HQ @CPU 2.60GHz 2.59GHz, 16GB RAM). The solution method is implemented in Java. The computational analysis was carried out using Python 3.10.

Figure 4 provides an overview of the convergence behaviour for all problem instances solved by Algorithm 3. In order to facilitate comparison between the different instances, the total costs obtained are normalized. Each figure contains data for the first 20.000 runs. For none of the instances solved a total run time of 10 h was exceeded. The results basically prove reasonable convergence behaviours towards the minimum total costs for all instances. However, particular differences between the instances can be observed. The lower the number of service trips, the faster the total costs obtained by Algorithm 3 decrease. It is noteworthy that the number of stop points has no visible influence on the speed of convergence.

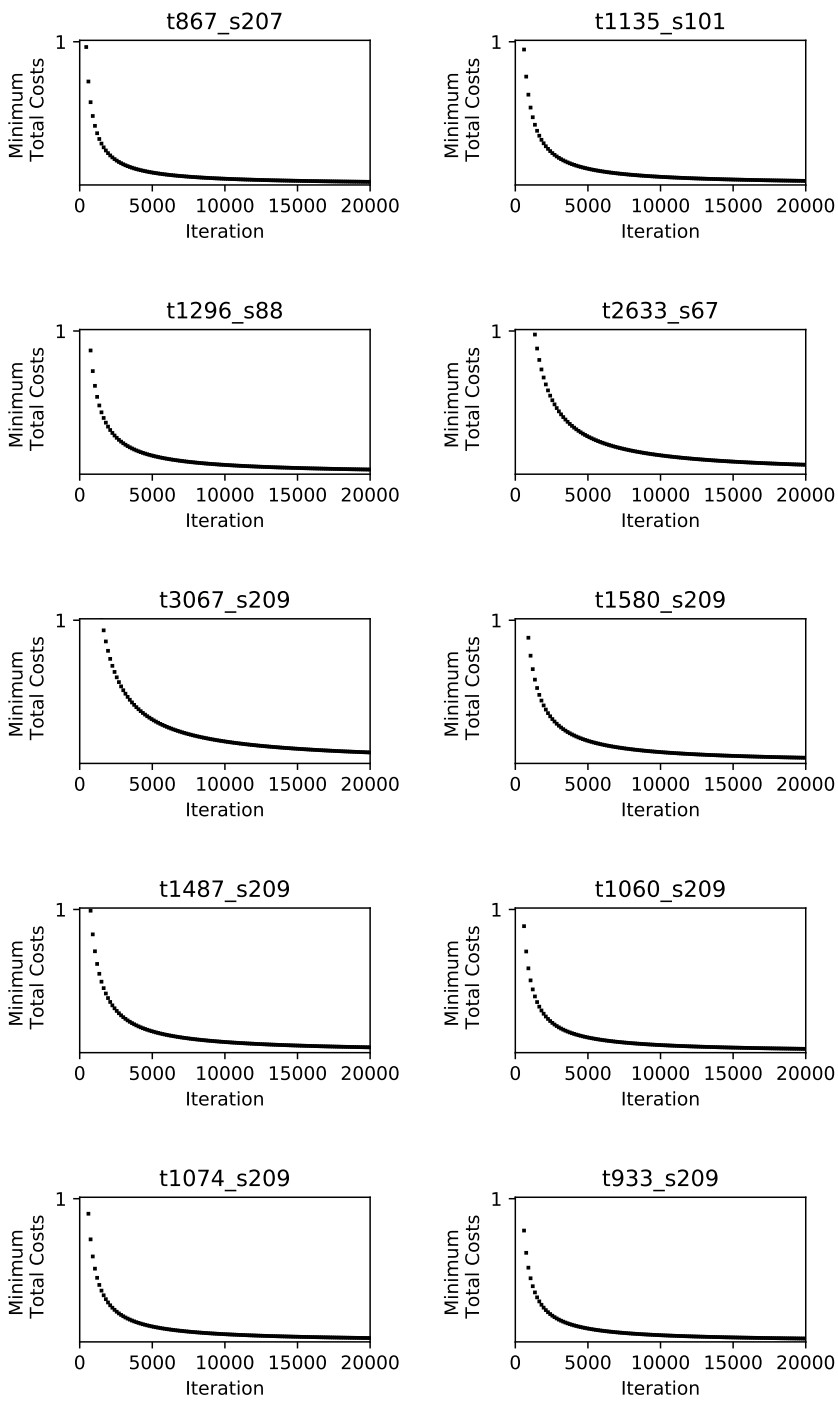

**Figure 4.** Overview of convergence behaviour for all problem instances solved by Algorithm 3.

## 6. Conclusions

We have introduced a novel solution method for simultaneous optimization of location planning of charging stations and vehicle scheduling for BEBs in public transport. To do so, we introduced the E-VSP-LP, which extends the standard E-VSP to incorporate location planning of charging stations. To solve the problem we developed a metaheuristic solution method based on VNS, as both problems are difficult to solve. To generate the necessary initial solutions we adapted the traditional savings algorithm. To evaluate the solution approach we performed a computational study based on real-world public transport data, with up to 3000 service trips and different battery capacities of the buses deployed. We also focused on a consideration of complete and partial battery charging procedures of the

batteries within vehicle rotations. In our study we compared the simultaneous solution approach to sequential planning to tackle the underlying problems.

Our experiments showed that simultaneous solving of location planning of charging stations and vehicle scheduling of BEBs is necessary as opposed to sequential planning. First, we demonstrated that sequential planning, first solving a standard VSP and afterwards a location planning problem for charging stations, generally leads to infeasible vehicle rotations for BEBs with regard to current battery technologies. Second, solving an E-VSP with given locations of charging stations entails significant increases in costs. Solving the E-VSP-LP, on the one hand, ensures the feasibility of the vehicle rotations. On the other hand, significantly lower total costs are achieved by comparison to solving an E-VSP, due to the higher degrees of freedom. This is particularly relevant for public transport companies that start operating electric bus fleets. With regard to complete and partial battery chargings, we found large cost savings in most cases when enabling partial chargings within the vehicle rotations.

Our paper can be extended by the following aspects. First, the proposed models do not deal with multiple depots. Incorporating this extension would most likely open up further potentials for cost savings, as already shown for the traditional VSP. Second, our solution method solves the E-VSP-LP heuristically. Exact solution approaches would be interesting for a better verification of the quality of heuristic solution methods. In addition, an interesting path for future research would be to develop additional algorithms for the generation of initial solutions as well as for improvement. Finally, more accurate models regarding the technical aspects of BEBs may be considered. It is conceivable to presume uncertain energy consumptions that may depend on weather conditions or the volume of traffic.

**Author Contributions:** Conceptualization, N.O. and N.K.; methodology, N.O. and N.K.; software, N.O. and N.K.; validation, N.O. and N.K.; formal analysis, N.O. and N.K.; investigation, N.O. and N.K.; resources, N.O. and N.K.; data curation, N.O. and N.K.; writing—original draft preparation, N.O. and N.K.; writing—review and editing, N.O. and N.K.; visualization, N.O. and N.K.; supervision, N.K.; project administration, N.K.; funding acquisition, N.K. All authors have read and agreed to the published version of the manuscript.

**Funding:** This work has been financially supported by the German Research Foundation [grant KL 2152/5-1]. The publication of this article was funded by Freie Universität Berlin.

**Institutional Review Board Statement:** Not applicable.

**Informed Consent Statement:** Not applicable.

**Data Availability Statement:** Not applicable.

**Conflicts of Interest:** The authors declare no conflict of interest.

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
