# Peer review of "Location Planning of Charging Stations for Electric Buses in Public Transport Considering Vehicle Scheduling: A Variable Neighborhood Search Based Approach"

_applsci, doi:10.3390/app12083855_

Round 1

Reviewer 1 Report

  1. I did notice several English errors. I suggest a full proofread and language check.
  2. Abstract could be shortened and be focused on the contribution of the study.
  3. Literature review could be improved. For instance, in Section 2.1, it seems like each study is reviewed in one paragraph (references [17] to [20]). These can be reorganised and revised in a more cohesive way.
  4. “Reference” column in Table 1 should have the full author(s) name and year.
  5. Some visualisation/examples could be added to Section 3 for a clearer understanding of the problem for the reader.
  6. In section 3, it is not clearly explained what decision variables are, and/or how an individual solution to the problem is represented.
  7. While the term “Vehicle Rotation” has been mentioned several times, no clear and concrete definition is provided (could go in Section 3).
  8. In Section 3, it should be justified how a long-term cost is added to the operational short-term cost in the objective function. Noting the first component is fixed for a number of years, whereas the second component could be variable/dynamic for real-life cases.
  9. It is not explained or justified why VNS (and not other metaheuristics) is selected for this study?
  10. Similarly, I think the proposed approach should be compared with other metaheuristics methods as well. Only comparing with the sequential planning is good but not enough to prove the merits of the study.

Reviewer 2 Report

The manuscript entitled “Location planning of charging stations for electric buses in public transport considering vehicle scheduling: a variable neighborhood search based approach” sent to me for review is interesting and has methodological and empirical values. The authors tackled extremely difficult problems. Due to the possible implications for practice, the article is worth publishing after adjusting it to the requirements.

The title corresponds to the content of the manuscript. The abstract, keywords, and introduction have been prepared properly. The methodical layer of the manuscript does not raise any objections, nor does the description of the results. However, I missed an extended discussion and associated conclusions.

I only have comments on the cosmetics of the manuscript (its editorial layer). They mainly concern mixing citation styles (MDPI style with footnotes - there should be no footnotes). The abbreviation "cf" should be omitted. Additionally, the references at the end of the article should be adapted to the requirements of the publishing house.

Some of the sources cited are incorrectly included (the reference in square brackets should be at the end of the sentence, not at the beginning). Change the syntax of the sentence so that there are no records, i.e.

"[12] consider a mixed fleet of vehicles consisting of electrically powered buses and buses without range limitations within the basic VSP". It is not legible; it introduces dissonance.

In addition, please give numbers to the formulas. A list of alphabets as tables, where the headings of all tables should be above the table, and below it should be informed about the source of the data (this was missing). The charts presented in Figure 4 are too small, it is difficult to notice the differences, maybe it would be worth proposing a different arrangement of the graphs in the magnification.

Tables 2-5 are improperly formatted, but also named (should be treated as an attachment since they appear after the reference list). Table titles are broken into multiple lines instead of one.

Reviewer 3 Report

 The paper proposes a variable neighborhood search (VNS)-based solution method for simultaneously optimizing charging station location planning and vehicle scheduling for battery-electric buses (BEB) in public transportation. The paper presents detailed results of solving the vehicle scheduling problem (VSP), electric vehicle scheduling problem (E-VSP), and charging station location planning (E-VSP-LP) based on real public transportation data and using a saving algorithm. The results show that the charging station location planning (E-VSP-LP) leads to a significant reduction in total cost compared to sequential planning, but several problems remain in the paper as follows.

  1. the unit of battery capacity in row 482 is generally used as Ah and the power consumed as kWh.
  2. The paper shows that the charging station location planning (E-VSP-LP) reduces the total cost, but there is no clear description of how much the reduction is.
  3. Figure 2 is not clearly presented. Figure 2 shows the distribution of cumulative service trips of the example over the course of a day. What do the authors illustrate by the cumulative service trips? How was it applied in the experiment? Figure 3 is vague as to the reason for choosing three examples.
  4. in the experimental design, only four groups of battery capacity are selected and only one model is considered, the experimental group data is small, and the reasons for selecting 60, 120, 300 and 500 kWh of battery capacity are not explained.
  5. The results are given in Tables II and III in lines 504 and 548, and the source process of the results is not stated.
  6. A series of comparisons are made in 5.4, and the comparisons can be illustrated in a table.
  7. What is the full name of BVNS in line 672 is not stated in the text.

Reviewer 4 Report

The paper presents a method for the simultaneous optimization of location planning of charging stations and vehicle scheduling for electric buses in public transport, using variable neighborhood search. It is shown that a simultaneous consideration of both problems is necessary in order to avoid cost increase.

The paper is well organized and well written.

The following aspects are recommended to be amended:

  • In section 5.4.2 it should be explained  how the specific instances have been selected, for which the results are presented (i.e. why t1060_s209, but not t1061_s224?).
  • The motivation why not a specific currency, but rather "monetary units" are used (although the cost values in the references are Euro or USD), should be provided in section 5.2.
  • In chapter 6, the conclusions could be extended to the brief elaboration of the significance of the research for practical purposes (i.e. from the viewpoint of a BEB operator).

Round 2

Reviewer 1 Report

Good work. The paper can now be accepted after addressing the following minor points:

  1. The format of 'URL references' (References 4, 5 and 9 in the bibliography section) should be double checked to comply with the journal style. 
  2. The URLs of both references #4 and #5 are 'dead links' as they took me to an 'access denied' and '404 not found' web pages.

Author Response

To the anonymous reviewer,

we thank you for your feedback and valuable comments which helped us again to improve the quality of our manuscript.

We revised the format of all references that contain an URL. We hope that the format now meets the requirements. Furthermore, we replaced one reference by a current reference and have ensured that all references are accessible.